# Two Congeneric Shrubs from the Atacama Desert Show Different Physiological Strategies That Improve Water Use Efficiency under a Simulated Heat Wave

**DOI:** 10.3390/plants12132464

**Published:** 2023-06-28

**Authors:** Enrique Ostria-Gallardo, Estrella Zúñiga-Contreras, Danny E. Carvajal, Teodoro Coba de La Peña, Ernesto Gianoli, Luisa Bascuñán-Godoy

**Affiliations:** 1Laboratory of Plant Physiology, Center of Advanced Studies in Arid Zones (CEAZA), La Serena 1700000, Chile; estrelladelpilar@gmail.com; 2Laboratory of Phytorremediation, Center of Advanced Studies in Arid Zones (CEAZA), La Serena 1700000, Chile; teodoro.cobadelapena@ceaza.cl; 3Laboratory of Plant Ecophysiology, Department of Biology, Universidad de La Serena, La Serena 1700000, Chile; dcarvajal@userena.cl; 4Instituto de Ecología y Biodiversidad (IEB), Santiago 8320000, Chile; 5Centro de Ciencia del Clima y la Resiliencia, CR2, Santiago 8320000, Chile; 6Laboratory of Functional Ecology, Department of Biology, Universidad de La Serena, La Serena 1700000, Chile; egianoli@userena.cl; 7Laboratory of Plant Physiology, Department of Botany, Universidad de Concepción, Concepción 4030000, Chile; lubascun@udec.cl

**Keywords:** *Atriplex*, Atacama Desert, C4 pathway, desert shrubs, heat wave, gas exchange, chlorophyll fluorescence

## Abstract

Desert shrubs are keystone species for plant diversity and ecosystem function. *Atriplex clivicola* and *Atriplex deserticola* (Amaranthaceae) are native shrubs from the Atacama Desert that show contrasting altitudinal distribution (*A. clivicola*: 0–700 m.a.s.l.; *A. deserticola*: 1500–3000 m.a.s.l.). Both species possess a C4 photosynthetic pathway and Kranz anatomy, traits adaptive to high temperatures. Historical records and projections for the near future show trends in increasing air temperature and frequency of heat wave events in these species’ habitats. Besides sharing a C4 pathway, it is not clear how their leaf-level physiological traits associated with photosynthesis and water relations respond to heat stress. We studied their physiological traits (gas exchange, chlorophyll fluorescence, water status) before and after a simulated heat wave (HW). Both species enhanced their intrinsic water use efficiency after HW but via different mechanisms. *A. clivicola*, which has a higher LMA than *A. deserticola*, enhances water saving by closing stomata and maintaining RWC (%) and leaf Ψ_md_ potential at similar values to those measured before HW. After HW, *A. deserticola* showed an increase of A_max_ without concurrent changes in g_s_ and a significant reduction of RWC and Ψ_md_. *A. deserticola* showed higher values of Chl*a* fluorescence after HW. Thus, under heat stress, *A. clivicola* maximizes water saving, whilst *A. deserticola* enhances its photosynthetic performance. These contrasting (eco)physiological strategies are consistent with the adaptation of each species to their local environmental conditions at different altitudes.

## 1. Introduction

Desert shrubs constitute a distinct physiognomic and ecological group with characteristic morphological and physiological adaptations [1,2]. Further, they often play a key role in the maintenance of plant diversity by acting as nurse plants [3] and may even impact ecosystem processes [4]. In the context of climate change and its profound effects on the physiology, phenology, abundance and distribution of species [5,6,7,8,9,10], it is essential to understand the mechanisms by which desert shrubs deal with climate change components.

Climate change involves not only increased temperature averages but also increased occurrence of climatic anomalies, such as heat waves [11]. In fact, heat waves are becoming increasingly frequent, more intense and broader in spatial extent [12,13,14]. Temperature controls the distribution, productivity, and physiological activity of plants, both at spatial and temporal scales [15,16]. Although they consist of relatively short events, heat waves may have significant ecological impacts and cause long-term ecological shifts [17,18]. Heat waves have been shown to affect plant physiological processes, such as the efficiency of photosystem II, stomatal conductance and leaf water potential [13,19,20]. Despite an increasing number of studies, the understanding of the effects of heat waves on plants is still fragmentary [13].

In eudicots, those with a C4 photosynthetic pathway are considered more tolerant to heat and water stress than C3 species because they have higher water-use efficiencies and negligible effects of heat on photorespiration [21,22]. Among predominantly C4 clades, the *Atriplex* genus (Amaranthaceae) is the largest one, including over 300 C4 species [23]. The *Atriplex* genus has spread worldwide, being mainly distributed across arid subtropical and temperate regions, particularly in harsh inland and coastal habitats [24]. Consequently, *Atriplex* species are usually labeled as highly tolerant plants against abiotic stresses, especially drought and salinity [25,26]. Adaptations to desert conditions in *Atriplex* shrubs leading to improved water economy have long been studied, including low mesophyll resistance and high stomatal resistance in *A. spongiosa* [27], steeply angled leaves during midday in *A. hymenelytra* [28], increased Na^+^ uptake in *A. halimus* [29], and increased osmotic adjustment in *A. nummularia* [30]. Furthermore, header books of plant (eco)physiology include *Atriplex* species when describing tolerance to high temperatures [31,32]. Nevertheless, little is known about their response to heat stress [33], much less about their response to heat waves.

Forty-six *Atriplex* species have been reported in Chile, most of them distributed in arid and semi-arid regions, which have experienced changes in the trends of heat waves and maximum temperatures [34,35]. During the 1961–2016 period, it was observed an increase in the number and duration of heat wave events [34]. Additionally, during the 1979–2015 period, there was an increase in maximum temperatures for summer and autumn, and high-altitude areas experienced a greater increase in maximum temperatures compared to lower and coastal areas [35]. Among the native *Atriplex* species inhabiting (semi)arid Chile, *Atriplex clivicola* and *A. deserticola* are two phylogenetically-close perennial endemic shrubs from the Atacama Desert [36,37]. These species share most of their latitudinal distribution but differ greatly in their altitudinal distribution [37]. *Atriplex clivicola* mainly inhabits coastal to lowland areas, from sea level up to 700 m a.s.l, whereas *A. deserticola* inhabits highlands from 1500 up to 3000 m a.s.l. Both species have a C4 photosynthetic pathway and Kranz anatomy, traits adaptive to high temperatures. However, besides sharing a C4 pathway, it is not clear how their leaf-level physiological traits associated with photosynthesis and water relations respond to heat stress. Here, we addressed whether *A. clivicola* and *A. deserticola* would enhance or maintain their leaf water status while maintaining carbon assimilation under a heat wave event. We evaluated gas exchange, chlorophyll *a* fluorescence, and leaf water status before and after a simulated heat wave. The overarching goal of this study was to study the strategies leading to water use efficiency in two representative desert shrubs under conditions of a heat wave, which is one of the components of current climate change.

## 2. Results

### 2.1. Chlorophyll a Fluorescence

The *Atriplex* species did not differ in the Fv/Fm parameter before or after HW. However, within species, Fv/Fm showed a significant reduction after HW yet maintained values above 0.7 (Figure 1A). The actual quantum yield (ΦPSII) only showed differences between species after HW, being lower in *A. clivicola* (Figure 1B). On the other hand, the quantum yield of energy dissipation (Fo/Fm) was higher in *A. deserticola* than in *A. clivicola* only before HW. The former reduced Fo/Fm significantly after HW, whereas similar values to those before HW were observed in *A. clivicola* (Figure 1C).

The components of NPQ, qE, qP and qL (at *p* < 0.05) (Figure 1D–F) showed differences between species after HW. After the HW simulation, the qE values were higher in *A. clivicola*, whilst qP and qL values were higher in *A. deserticola*.

No significant changes in ETR were observed within species before or after HW (Figure 1G). We did find differences in ETR between species, but only after HW, being lower for *A. clivicola*.

### 2.2. Gas Exchange Variables

The photosynthetic rate (A_max_) values showed differences between species only before HW (Figure 2A), being ~47% higher in *A. clivicola* (14.32 µmol CO_2_ m^−2^ s^−1^) than in *A. deserticola* (9.77 µmol CO_2_ m^−2^ s^−1^). After HW, the photosynthetic rate of *A. clivicola* remained similar to values observed before HW. However, the photosynthetic rate of *A. deserticola* increased by 39% after HW.

Stomatal conductance (g_s_) was higher in *A. clivicola* than in *A. deserticola* before HW, whilst no difference between species was observed after HW (Figure 2B). *A. deserticola* showed similar values of g_s_ before and after HW (0.21 and 0.22 mol H_2_O m^−2^ s^−1^, respectively). On the other hand, a significant decrease in g_s_ (~43%) was observed in *A. clivicola* after HW, reaching similar values to those of *A. deserticola*.

Leaf transpiration ra©(E) showed no differences between species either before or after HW (Figure 2C). Within species, *A. clivicola* showed a ~32% decrease in E after HW (*p* = 0.0506), whereas the E of *A. deserticola* was not affected by HW.

The intrinsic water use efficiency (iWUE) was remarkably similar in *A. clivicola* and *A. deserticola* both before and after HW (Figure 3D). Within species, after HW, a significant increase of ca. 39 and 81% in iWUE was observed in *A. clivicola* and *A. deserticola*, respectively (Figure 2D).

### 2.3. Photosynthetic Temperature–Response Curves

The optimum photosynthetic temperature (T_opt_) values differed between species (*p* < 0.05) (Table 1). *A. deserticola* averaged an optimum temperature of 29.63 °C, while *A. clivicola* had an optimum temperature of 27.07 °C. Similarly, the photosynthetic rate at thermal optimum (A_opt_) was higher (*p* < 0.05) in *A. deserticola* (19.64 µmol CO_2_ m^−2^ s^−1^) than in *A. clivicola* (9.30 µmol CO_2_ m^−2^ s^−1^).

The thermal maximum value where carbon gain reach zero (T_max_) was higher (*p* = 0.0538) in *A. deserticola* (46.01 °C) compared to *A. clivicola* (36.67 °C). Interestingly, the thermal breadth (T_br_), which reflects the potential increase in T_max_ with no changes in T_opt_ was similar in both species (Table 1).

### 2.4. Leaf Mass Per Area and Water Relation Variables

Statistically significant differences (*p* < 0.05) in leaf mass per area (LMA) between species were found, where *A. clivicola* showed higher LMA values (131.95 g m^−2^) than *A. deserticola* (105.01 g m^−2^) (Figure 3A). Specifically, both species showed similar values of leaf area, the differences in LMA lying on the greater leaf mass of *A. clivicola*.

Leaf relative water content (RWC) differed between species after HW (*p* < 0.05), being lower in *A. deserticola* (64.4%) than in *A. clivicola* (70.58%) (Figure 3B). RWC was affected in *A. deserticola* after HW, with a decrease of ca. 7%, whereas no change was observed in *A. clivicola*.

Both species showed similar values of leaf water potential at midday (Ψ_md_) before HW (Figure 3C). However, after HW leaf Ψ_md_ values were lower in *A. deserticola* (−2.49 MPa). Specifically, *A. deserticola* showed a significant decrease of leaf Ψ_md_ of ca. 14% after HW, while leaf Ψ_md_ of *A. clivicola* was not affected by HW.

## 3. Discussion

In this study, we examined photosynthetic and water relation traits of two *Atriplex* species exposed to a three-day simulated heat wave (HW). In general, photochemistry, carbon metabolism and hydraulics are the processes mostly affected by heat stress during the vegetative phase of plants [12]. Overall, both *Atriplex* species were resistant to high temperatures, an expected feature for C4 plants, which have evolved in warmer and dry environments [17,23]. Nevertheless, the HW had distinctive effects on the photosynthetic and water relation traits of *A. clivicola* and *A. deserticola*, suggesting species-specific mechanisms underlying its remarkable improvement in iWUE. Specifically, the response to the HW of *A. clivicola* was to enhance water saving, whereas the response of *A. deserticola* was to enhance its photosynthetic performance.

Heat stress, alone or combined with other abiotic stress, such as drought, can severely affect the integrity of thylakoid membranes and the function of PSII [38,39,40]. Our results showed that after HW, there was a reduction in the maximum quantum efficiency of PSII (Fv/Fm) in both *Atriplex* species, although with average values no lower than 0.74 (Figure 1A). These results reflect that the maximum quantum yield capacity of photosystem II was not severely affected by the heat stress in either species. By contrast, the actual quantum yield (ΦPSII) was more sensitive to heat stress, particularly in *A. deserticola*. It has been reported that negative effects on photochemistry and light-energy partition, particularly on F_v_/F_m_ and ΦPSII, are apparently temporal, and recovery may occur rapidly in sensitive plants, such as soybean, whereas even positive effects have been observed in heat-tolerant plants [38,41]. Notably, we observed for both *Atriplex* species an increase in F_0_ after the heat wave, but contrary to what is expected; it was accompanied by an increase in F_m_ [42]. Indeed, a higher increase of F_m_ in *A. deserticola* would explain the reduction of its quantum yield of energy dissipation (F_0_/F_m_) and sustain its actual quantum yield after HW. Together, these results suggest that, after three days of heat stress, the integrity of thylakoid membranes would be barely impaired, and the photosynthetic machinery remains functional.

Here, the values obtained for NPQ components qE, qP and qL reveal different mechanisms in the *Atriplex* species to deal with light energy after HW (Figure 1D−F). Specifically, the higher values of q_E_ in *A. clivicola* suggest that HW would affect the capability of the photochemical apparatus to use light energy in photochemistry and must activate the components that rapidly relax high-energy state quenching in order to drain the excess of light energy and avoid potential photoinhibition and photo-oxidative stress [43,44,45]. On the other hand, HW did not alter the photosystem functioning of *A. deserticola*, which maintained qP and qL values without significant variations compared to those observed before HW and were comparatively higher than those of *A. clivicola* after HW. Since these two components reflect the partition of light energy into photochemistry (qP) and the fraction of open PSII centers with oxidized QA (qL), we can infer that the thylakoid membranes and proteins of *A. deserticola* remain stable and functional after three days of high temperature. The latter would support the observation that ETR did not change in *A. deserticola* with HW and was higher than *A. clivicola* after heat stress. Similar results have been reported in high-temperature resistant cultivars of wheat and quinoa [38,46], which suggests that the ability to increase chlorophyll content under heat stress is pivotal to diminishing heat damage of thylakoid membranes and enhances the efficiency of electron transport at high temperatures. Although here, we did not measure chlorophyll content; there was no visual evidence of chlorosis during and after HW. In sum, a hypothesis derived from our results is that, under heat stress, *A. deserticola* would either increase chlorophyll content or show a rapid turnover of chlorophyll, whereas *A. clivicola* would use photoprotective mechanisms mainly associated with the xanthophyll cycle to deal with excess light energy.

It is considered that extreme supra-optimal temperatures usually restrict photosynthesis by affecting chloroplasts membrane fluidity and protein stability, enzyme properties, and decreasing CO_2_ solubility and stomatal aperture, with a further decrease of CO_2_ flux [22,41,47]. Again, the results of temperature–response curves and gas exchange support the notion that *A. clivicola* and *A. deserticola* are high-temperature resistant plants. Despite differences between both *Atriplex* species, they have high photosynthetic optimum temperatures compared to other woody plant species [48]. The maximum photosynthetic capacity (A_max_) of *A. clivicola* was not affected by HW despite a significant reduction in stomatal conductance (g_s_), whereas an increase of A_max_ was observed in *A. deserticola* without evident changes in g_s_. Both species reduced the apparent transpiration rate (E) while significantly enhancing their intrinsic water use efficiency (iWUE). Similar results have been reported in drought-tolerant crop varieties [39]. Additionally, Eustis et al. [38] found that heat stress alone increased carbon assimilation in several quinoa genotypes but, contrary to our findings, at the expense of higher stomatal conductance and plant water demand. Heat-induced limiting carbon assimilation is mainly due to the impairment of electron transport and Rubisco activase capacity [15]. Hence, in the case of *A. deserticola*, the increase in A_max_ can be partly explained by the ability to maintain the efficiency of the photochemical process and ETR after HW, as mentioned above. The fact that both species had high LMA values can be related to our observed A_max_ values since leaf size and thickness are key traits in the response of leaf photosynthesis to heat and thermal sensitivity [13,32,49]. Indeed, in a Mediterranean sclerophyllous ecosystem, plants with thicker leaves were less affected by a two-days heat wave (>45 °C) than plants with larger and thinner leaves [50].

The evaporative demand of water increases with temperature; therefore, leaf hydraulic responses are important to cope with the increase of leaf-to-air vapor pressure deficit [51]. Coordination between leaf hydraulics and leaf gas exchange is key for the balance of carbon gain and water loss [52,53]. Here, the relative water content (% RWC) and leaf water potential (Ψ_md_) of *A. clivicola* were insensitive to heat waves, whereas *A. deserticola* displayed a significant decrease in both parameters (Figure 3B,C). Given that the former showed a reduction of g_s_ after HW, this helps the plant decrease leaf evaporative water loss and would partly explain the RWC values observed after the heat stress, which in turn would contribute to keeping leaf water potential. On the other hand, the fact that g_s_ of *A. deserticola* was insensitive to HW, a higher evaporative water loss would be a plausible explanation for the reduction in RWC and Ψ_md_. However, given that A_max_ increased while E decreased compared to the values before HW, leaf water loss would be within a tolerable margin that would not jeopardize carbon assimilation and water use efficiency.

Environmental gradients shape the expression of ecophysiological traits [54,55]. Our results can be associated with the local environmental conditions within the species’ altitudinal distribution. *A. clivicola* is a coastal lowland species, whereas *A. deserticola* inhabits high-altitude ecosystems of the Andean range. This distribution imposes marked differences regarding microclimatic conditions, such as thermal amplitude, solar radiation and air humidity. The zone in which *A. clivicola* is distributed has a strong ocean influence, low thermal oscillation, and there is an important contribution of fog as a source of water for plants [56,57,58]. Therefore, avoidance of water loss under a heat wave is to be expected. On the other hand, *A. deserticola* must face high thermal amplitude within a single day, along with high irradiation [59]. Additionally, in high-mountain habitats, low partial pressure of gases and windy days may affect CO_2_ availability [60]. Thus, it is reasonable to expect that *A. deserticola* would show better photosynthetic performance under heat stress. Therefore, the contrasting ecophysiological strategies exhibited by *A. clivicola* and *A. deserticola* are consistent with the adaptation of each species to their local environmental conditions at different altitudes. Notably, despite the above-mentioned differences, both species coordinate their photosynthetic and water relation traits to enhance water use efficiency under heat stress.

## 4. Materials and Methods

### 4.1. Plant Material and Growth Conditions

One-year-old plants of *A. clivicola* and *A. deserticola* were obtained from the nursery garden of the Chilean Forestry Institute (INFOR-Sede Diaguita, La Serena, Chile). Plants were propagated vegetatively and, once rooted, placed on individual plastic bags of 20 cm × 20 cm filled with organic potting mix. The plants were watered thrice a week during spring and summer and twice a week during autumn–winter. Once well established, during spring, a total of 60 plants (*n* = 30 per species) were transferred to a greenhouse at Universidad de La Serena, Campus Andres Bello (29°54′53.04″ S 71°14′31.22″ W). Plants were watered at field capacity thrice a week, based on the values obtained by a Time Domain Reflectometer soil moisture meter TDR350 (FieldScout Spectrum Technologies, Inc., Aurora, IL, USA). The greenhouse roof was covered by a white polyethylene sheet providing ca. 800–1200 µmol photons m^−2^ s^−1^ at solar midday. All ecophysiological traits described below were evaluated before and after the simulated heat wave.

### 4.2. Heat Wave Simulation

We selected twelve healthy plants per species in order to submit them to a three-day simulated heat wave. Plants were transferred to a growth chamber of 12 m^2^ with controlled temperature and light conditions. The temperature control consists of a cooling system of a 2 hp condensing unit coupled with heat convection plates controlled by the software Sitrad Pro (www.sitrad.com.br, accessed on 24 November 2022, Full Gauge Controls, Canoas, RS, Brazil). The light was provided by 1000 W RGB LED panels. Plants were preconditioned to the chamber conditions under 25 °C and 16/8 h day/night during a week. Plants were placed on metal racks, each with three LED panels 50 cm above the top of the plants, to ensure a homogeneous light distribution. The PAR (photosynthetically active radiation) provided by the led panels was of ca. 400 µmol photons m^−2^ s^−1^. Watering was at field capacity, as described above. After the preconditioning period, the chamber temperature was set to 28 °C night, 33 °C from 6 to 11 am, and 38 °C during the rest of the day (Appendix A), according to the 16/8 h day/night photoperiod indicated above. To define the heat wave conditions, we analyzed 20 years of data on the absolute maximum temperature during spring–summer (CEAZA meteorological stations network; www.ceazamet.cl, accessed on 24 November 2022). The 3 d duration of the simulated heat wave followed the IPCC definition [61].

### 4.3. Chlorophyll Fluorescence and Gas Exchange

Chlorophyll fluorescence measurements were conducted with an FMS2 pulse-modulated fluorometer (Hansatech Instruments Ltd., Norfolk, UK) to determine the dark-adapted parameters and NPQ components. The actinic light used was 400 µmol quanta m^−2^ s^−1^. Forty-eight leaves (two leaves per individual, a total of 24 individuals) were dark-adapted overnight prior to measurements. The maximum and actual quantum yield of PSII (Fv/Fm and ΦPSII), the quantum yield of energy dissipation (Fo/Fm), non-photochemical quenching (NPQ) and the electron transport rate (ETR) were calculated as described in Maxwell and Johnson [62,63]. NPQ components (qE, qI and qL) were calculated from Fm’ dark relaxation kinetics after high light exposure for 1 h, as described by Bravo et al. [64] and Bascuñán-Godoy et al. [65] with slight modifications.

Leaf gas exchange measurements were conducted using a gas exchange system (LI-6400, Li-Cor Inc., Lincoln, NE, USA) with a leaf chamber of 2 cm^2^ with a LED light source (LI-6400-40). We conducted light response curves on three individuals per species to determine the light saturation point for both *Atriplex* species. CO_2_ concentration was set at 400 ppm, and nine consecutive light steps of 10 min each to ensure CO_2_ assimilation was stable were set as follows: 0, 50, 100, 300, 500, 900, 1200, 1500 and 2000 µmol photons m^−2^ s^−1^ [66]. Leaves were carefully placed in the sensor head, ensuring contact with the leaf thermocouple. A picture of each leaf inside the chamber’s gasket was taken to determine the leaf area and subsequently use it to calculate gas exchange parameters. Light-saturated CO_2_ assimilation (A_max_), stomatal conductance (g_s_), and apparent transpiration rate (E) were measured in twelve individuals per species from 9:00 to 13:00. Gas exchange parameters were recorded 10 min after clamping the leaf. Leaf chamber conditions were set at 400 ppm of CO_2_, 1200 µmol photons m^−2^ s^−1^ (90:10% red: blue light), 60–65% relative humidity and 25 °C block temperature. For each gas exchange measurement, the intrinsic water use efficiency (iWUE) was calculated as the ratio of photosynthesis (A_max_) over stomatal conductance (g_s_).

We conducted leaf temperature response curves to determine the optimum photosynthetic temperature (T_opt_), the temperature compensation point (T_max_), and the photosynthetic thermal breadth (T_br_). We used five individuals of each *Atriplex* species for measurements of net photosynthesis temperature (A-T) response curves (Appendix A). A-T curves were performed between 09:00 to 14:00 h on healthy, fully expanded leaves using an open gas exchange system (Li-6400XT, Li-Cor Inc., Lincoln, NE, USA), equipped with a water bath and a temperature expansion kit to reach both lowest (<15 °C) and highest temperatures (>36 °C), set at 1500 µmol photons m^−2^ s^−1^ (10% blue), an ambient CO_2_ concentration of 400 mmol CO_2_ mol^−1^, and with 12 different block temperatures (12 °C, 15 °C, 18 °C, 21 °C, 24 °C, 27 °C, 30 °C, 33 °C, 36 °C, 40 °C, 45 °C and, 50 °C). The relative humidity (RH) of the sample was regulated to 50 ± 0.5% during measurements except at block temperatures >36 °C, where it was allowed to drop to avoid condensation damage within the IRGA [67]. Once parameters were at steady-state for at least 6 min for each block temperature, we recorded leaf temperature (T_leaf_) and photosynthetic rate (A_net_). Then, to determine the temperature at the carbon compensation point (T_max_), the temperature at optimum photosynthesis (T_opt_), the photosynthetic rate at optimum temperature (A_opt_), and the breadth of temperature optimum (T_span_), we fit the data using a standard quadratic equation [67] as A_net_ = aT_leaf_^2^ + bT_leaf_ + c, where A_net_ is net photosynthesis (μmol m^−2^ s^−1^) at T_leaf_ (°C), and a, b and c are coefficients that describe the A-T response. Curves were fitted using linear models with quadratic components (Yi = β0 + *β*_1_X_i_ + *β*_2_X_i_^2^ + ε_i_), using T_leaf_ as the independent variable (X_i_) and A_net_ as the dependent variable using the “lm” function in the ‘stats’ package in R version 4.0.0 [68].

### 4.4. Leaf Mass Per Area, Relative Water Content and Leaf Water Potential

Leaf samples employed for gas exchange and chlorophyll fluorescence measurements (24 samples per species) were used to determine the dry leaf mass per unit area (LMA) and relative water content (RWC %). LMA was calculated as the ratio between the dry leaf weight and leaf area obtained through the ImageJ software version 1.54e (ImageJ; nih.gov, accessed on 23 November 2022), according to Perez-Harguindeguy et al. ([69]). RWC of each leaf was determined as in Ostria-Gallardo et al. [70]: % RWC = ((Fw − Dw)/(Tw − Dw)) × 100; were Fw = fresh weight, Dw = dry weight, and Tw = turgid weight. The turgid weight was obtained by dipping the leaf into a 2 mL Eppendorf tube full of distilled water and storing it at 4 °C for 24 h prior to weighing.

For leaf water potential (Ψ_md_, MPa) at midday (11:00–14:00), a healthy apical branch of each individual (*n* = 24) was detached using a fresh razor blade, immediately wrapping the cut end with a wet paper and storing it in a sealed dark plastic bag with a moist paper towel. We measured Ψ_md_ (MPa) using a Scholander-type pressure chamber (PMS Instrument Company, Corvallis, OR, USA) 10–45 min after sample collection.

### 4.5. Data Analysis

Data were checked for normality assumptions and variance homoscedasticity with the InfoStat software [71]. Accordingly, the parametric one-way ANOVA or non-parametric Kruskal–Wallis analyses were used to compare means within species and between species separately. When significant differences were found, we used post hoc tests with Tukey or pairwise comparisons (*p* ≤ 0.05), depending on the parametric or non-parametric nature of the data.

## 5. Conclusions

Based on their photosynthetic and water relation traits, both *Atriplex* species enhanced their water-use efficiencies after a three-days heat wave treatment (HW). This enhancement was governed by different mechanisms depending on the species. *A. clivicola* achieved it by enhancing water saving by closing stomata and maintaining the leaf RWC (%) and Ψ_md_ at similar values to those registered before HW. By contrast, *A. deserticola* enhanced its photosynthetic performance at the expense of water loss at the foliar level. Specifically, after HW, *A. deserticola* showed an increase of Amax without evident changes in stomatal conductance and with higher values of chlorophyll fluorescence parameters associated with photochemistry and electron transport.

It is expected that heat waves and other extreme climatic events associated with climate change will become more frequent and intense. Here we provide evidence of different ecophysiological strategies to cope with heat waves in desert shrubs, which are keystone species for plant diversity and ecosystem function, and hence it is essential to understand the mechanisms by which they deal with climate change.

## Figures and Tables

**Figure 1 plants-12-02464-f001:**
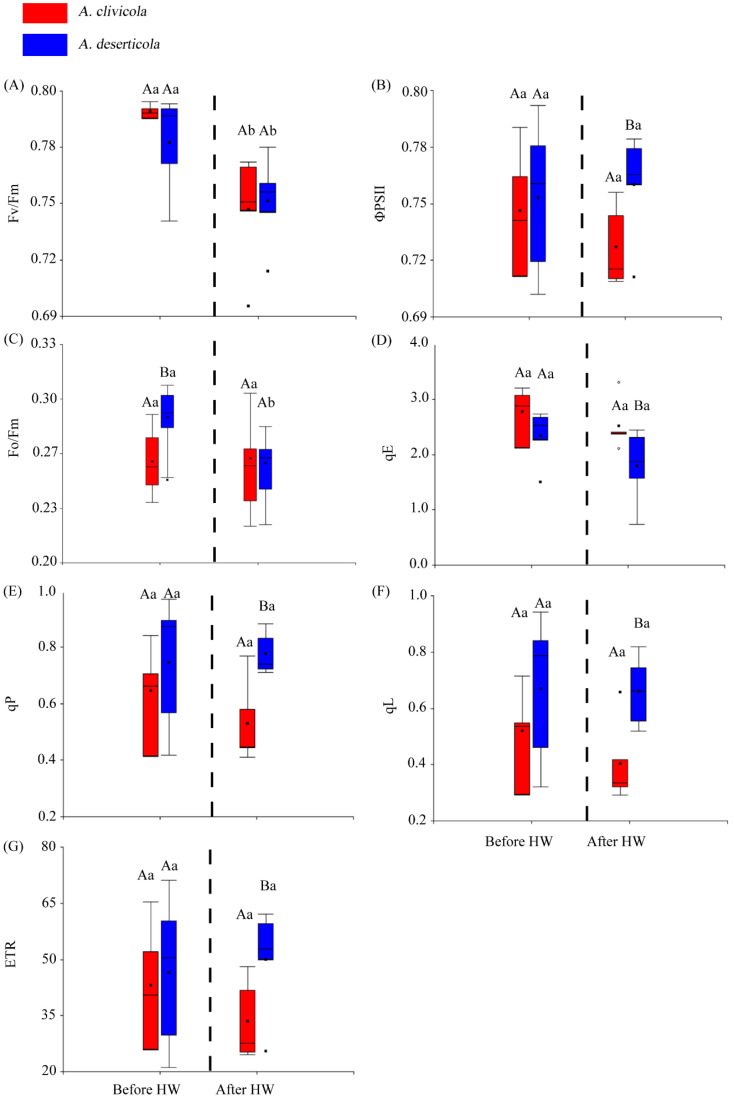
Box plots for the chlorophyll *a* fluorescence parameters of *Atriplex clivicola* (red; *n* = 12 individuals) and *Atriplex deserticola* (blue; *n* = 12 individuals) species before and after the simulated heat wave (HW). The dashed line of each plot separates the values obtained before and after HW. The black dot and horizontal line inside each box indicates the mean and median values. Vertical lines outside the box indicate the extreme lower and upper values of the dataset. Upper case letters indicate significant differences between species before and after HW. Lowercase letters indicate significant differences within species when values before and after HW are compared. Fv/Fm Maximum quantum efficiency of photosystem II (**A**); ΦPSII Actual quantum yield of photosystem II (**B**); Fo/Fm Quantum yield of energy dissipation at time = 0 (**C**); qE Quenching component associated to energy dissipation (**D**); qP Quenching component associated to photochemistry (**E**); qL Fraction of open PSII centers (with QA oxidized) on the basis of a lake model for the PSII photosynthetic apparatus (**F**); ETR Electron transport rate (**G**).

**Figure 2 plants-12-02464-f002:**
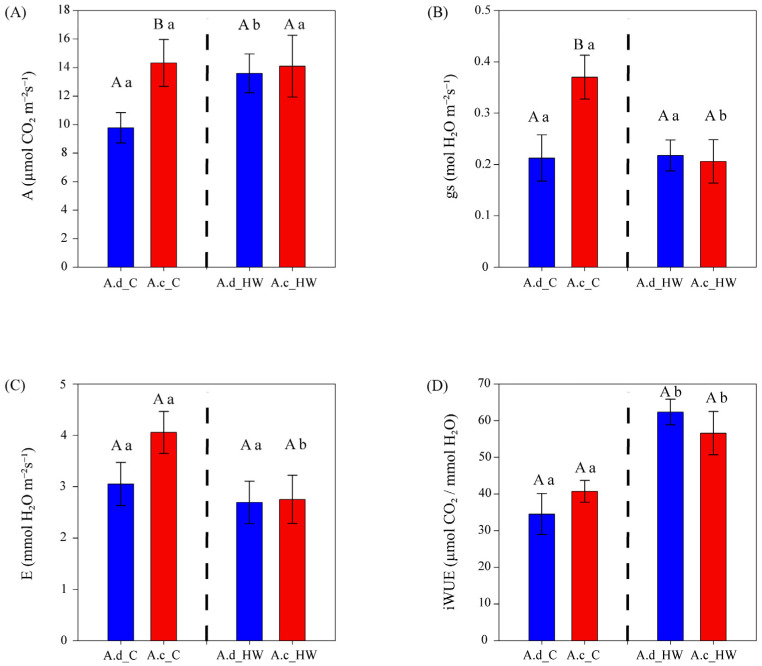
Gas exchange parameters of *A. clivicola* (red; *n* = 12 individuals) and *A. deserticola* (blue; *n* = 12 individuals) before and after the simulated heat wave (HW). The dashed line inside each plot separates the values obtained before and after HW. Bars and vertical lines above the bars indicate the mean and the standard error values, respectively. Upper case letters indicate significant differences between species either before HW or after HW. Lowercase letters indicate significant differences within species when values before and after HW are compared. A Photosynthetic rate at light saturation (**A**); gs Stomatal conductance (**B**); E Transpiration rate (**C**); iWUE Intrinsic water use efficiency (**D**).

**Figure 3 plants-12-02464-f003:**
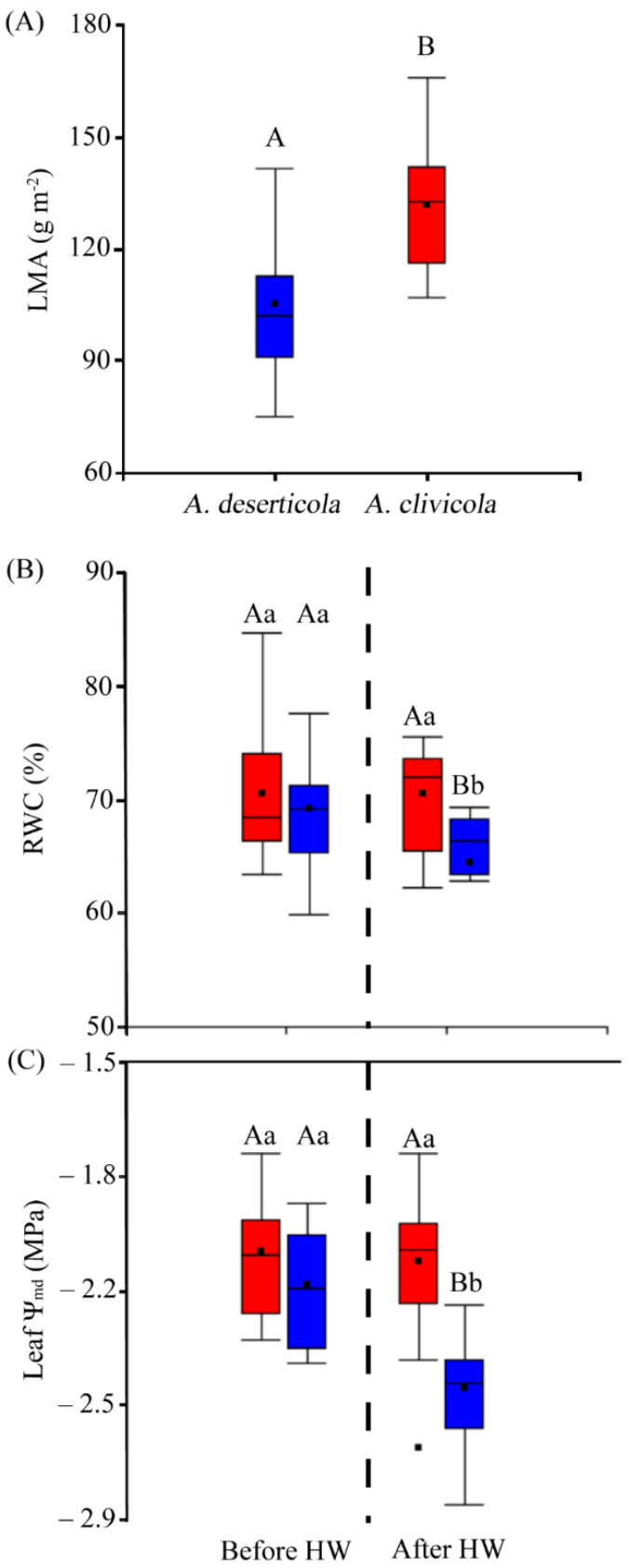
(**A**) Leaf mass per area (LMA) values of *A. deserticola* (*n* = 12 individuals) and *A. clivicola* (*n* = 12 individuals). (**B**) The percentage of relative water content (RWC) of *A. clivicola* and *A. deserticola* before and after simulated heat wave (HW). (**C**) Leaf water potential at midday (Ψ_md_) of *A. clivicola* and *A. deserticola* before and after the simulated HW. The dashed line inside each plot separates the values obtained before and after HW. The black dot and horizontal line inside the box plots indicates the mean and median values. Vertical lines outside the boxes indicate the extreme lower and upper values of the dataset. Upper case letters indicate significant differences between species either before HW or after HW. Lowercase letters indicate significant differences within species when values before and after HW are compared.

**Table 1 plants-12-02464-t001:** Parameters from the photosynthetic temperature–response curves of *A. clivicola* (*n* = 4 individuals) and *A. deserticola* (*n* = 4 individuals). The table depicts the thermal sensitivity of photosynthesis to high temperatures. The values correspond to the average and standard error for the maximum temperature at the carbon compensation point (T_max_), the optimum temperature for photosynthesis (T_opt_), the photosynthetic rate at optimum temperature (A_opt_), and the thermal breadth (T_br_).

Species	T_max_ (°C)		T_opt_ (°C)		A_opt_ (µmol CO_2_ m^−2^ s^−1^)		T_br_ (°C)	
*A. clivicola*	39.67	±1.65	27.05	±0.85 *	9.30	±0.81 *	11.28	±0.90
*A. deserticola*	46.01	±2.07	29.63	±0.62 *	19.64	±2.54 *	14.65	±1.50

Asterisk denotes significant difference between species at *p* < 0.05.

## Data Availability

Data are available on request from the authors.

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
