# Peer review of "Two Congeneric Shrubs from the Atacama Desert Show Different Physiological Strategies That Improve Water Use Efficiency under a Simulated Heat Wave"

_plants, 2023, doi:10.3390/plants12132464_

Round 1
Reviewer 1 Report
Dear authors
Please take into consideration the corrections, suggestions and comments on the attached file.
Kind regards

Author Response
Dear Reviewer 1,
We truly appreciate your time dedicated to read our manuscript as well as all the comments and suggestions. These have helped us to improve our manuscript.
We have incorporated all the changes and corrections related to format, such as italicize scientific names, text format, acronyms, and units. That was mainly a format mistake when transferring the paragraphs to the template document. We have also changed the sentence of line 98-99 to a less generic sentence. We have added the n number of replicates to all the figures and table.
Regarding the specific question about the actinic light used in chlorophyll fluorescence measurement in line 341, it was a typo mistake and was corrected in the main manuscript. We have also change "fluorescence chamber" in line 351 by "leaf chamber of 2 cm2 with a LED light source". Regarding the question in line 354, about the waiting time for stabilization of CO2 assimilation, there was a mistake since we write the waiting time of other of our experiments conducted in crops, that stabilize within 2-3 min. In the specific case of the Atriplex species studied here, both stabilize within 8-10 min during a preliminary screening. Thus, we now indicate in the corrected manuscript that the waiting time before recording CO2 assimilation was of 10 min.
Reviewer 2 Report
The manuscript is well structured and written. The M&M are well explained and detailed. Results and discussion are coherent and accurately presented and discussed. The introduction is clear and well-focused on the objectives.
The Authors referred to instantaneous WUE (defined as A/E) in M&M, results and graph 2, but at line 260 (discussion) they referred to intrinsic WUE (A/gs). So please, be consistent.
Moreover, I would strongly suggest to use intrinsic WUE instead of instantaneous WUE, for several reasons:
(i) stomatal conductance is a variable directly measured within Li-6400 leaf chamber, while E is only a calculated variable, which is more affected by microenvironment modifications (such as temperature, relative humidity);
(ii) differences in A/gs among species/genotypes indicate a genetic basis for this ratio;
(iii) there is a significant diurnal time effect on A and E, further than a seasonal variation along the season because of the changing environmental conditions and the physiological changes (e.g. leaf aging), which affect leaf photosynthesis and transpiration.
For more details see:
Medrano, H., Tomás, M., Martorell, S., Flexas, J., Hernández, E., Rosselló, J., ... & Bota, J. (2015). From leaf to whole-plant water use efficiency (WUE) in complex canopies: Limitations of leaf WUE as a selection target. The Crop Journal, 3(3), 220-228.
H. Medrano, J. Gulías, M. Chaves, J. Galmés, J. Flexas, Photosynthesis water-use efficiency, in: J. Flexas, F. Loreto, H. Medrano (Eds.), Terrestrial Photosynthesis in a Changing Environment, A Molecular, Physiological and Ecological ApproachCambridge University Press, Cambridge 2012, pp. 529–543.
H. Medrano, J. Flexas, M. Ribas-Carbó, J. Gulías, Measuring water use efficiency in grapevines, in: S. Delrot, H. Medrano, E. Or, L. Bavaresco, S. Grando (Eds.), Methodologies and Results Grapevine Research, Springer, Germany 2010, pp. 57–60.
Author Response
Dear Reviewer 2,
We truly appreciate the time for reading our manuscript and for your recommendations. Your feedback have helped us to improve our manuscript.
We have corrected the inconsistencies pointed out by the reviewer, and we have also changed the use of instantaneous water use efficiency by the intrinsic use efficiency. We agreed with the rationale indicated by the reviewer, so now the M&M, results, and discussion about WUE is based on the intrinsic water use efficiency.